# RNA Epigenetics: Fine-Tuning Chromatin Plasticity and Transcriptional Regulation, and the Implications in Human Diseases

**DOI:** 10.3390/genes12050627

**Published:** 2021-04-22

**Authors:** Amber Willbanks, Shaun Wood, Jason X. Cheng

**Affiliations:** Department of Pathology, Hematopathology Section, University of Chicago, Chicago, IL 60637, USA; awillbanks@bsd.uchicago.edu (A.W.); shaunwood@bsd.uchicago.edu (S.W.)

**Keywords:** 5’ cap (5’ cap), 7-methylguanosine (m^7^G), R-loops, N6-methyladenosine (m^6^A), RNA editing, A-to-I, C-to-U, 2’-O-methylation (Nm), 5-methylcytosine (m^5^C), NOL1/NOP2/sun domain (NSUN), MYC

## Abstract

Chromatin structure plays an essential role in eukaryotic gene expression and cell identity. Traditionally, DNA and histone modifications have been the focus of chromatin regulation; however, recent molecular and imaging studies have revealed an intimate connection between RNA epigenetics and chromatin structure. Accumulating evidence suggests that RNA serves as the interplay between chromatin and the transcription and splicing machineries within the cell. Additionally, epigenetic modifications of nascent RNAs fine-tune these interactions to regulate gene expression at the co- and post-transcriptional levels in normal cell development and human diseases. This review will provide an overview of recent advances in the emerging field of RNA epigenetics, specifically the role of RNA modifications and RNA modifying proteins in chromatin remodeling, transcription activation and RNA processing, as well as translational implications in human diseases.

## 1. Introduction

In eukaryotes, genomic DNA and its associated proteins, primarily histones, are organized into chromatin structural domains that affect the binding of transcription factors (TFs), recruitment of RNA polymerases (RNAPs) and the initiation and elongation of transcription [1]. The advent of new genome-wide sequencing technologies has identified chromatin-associated RNAs (ChrRNAs) [2], including nascent RNAs, coding RNAs and various regulatory non-coding RNAs (ncRNAs). RNAs can act either *in*
*cis* by targeting genomic DNA or *in*
*trans* through RNAP complexes and interactions with DNA-binding proteins, such as TFs and DNA/histone modifiers, to regulate chromatin structure and gene expression. There are extensive studies and reviews of the roles of ncRNAs in the regulation of DNA/histone modifications, chromatin structure, and gene expression [3,4,5,6,7,8,9,10]. In contrast, the role of RNA modifications and RNA modifying proteins (RMPs) in chromatin plasticity and transcription regulation remains largely unknown. This review will discuss recent studies, which suggest RNA modifications and RMPs function to fine-tune chromatin structure, in turn facilitating transcription activation or repression.

## 2. Co- and Post-Transcriptional RNA Modifications and RNA Modifying Proteins

### 2.1. Overview of RNA Modifications and RMPs

Over 170 chemical modifications have been identified in RNA [11]. The associated RNA modifying proteins (RMPs) are classified into three groups based on their roles in RNA modification: (1) writers: enzymes that catalyze specific RNA modifications, (2) readers: enzymes that recognize and selectively bind to RNA modifications and (3) erasers: enzymes that remove specific RNA modifications. These RMPs are distributed throughout the nuclei, cytoplasm and mitochondria and are involved in nearly all essential bioprocesses (Figure 1) [11,12,13]. In this section, we will provide a brief update on the common RNA modifications and their associated RMPs as well as their impact on chromatin structure and transcription regulation. The common RNA modifications and their writers, readers and erasers are shown in Figure 2 and Figure 3; N6-methyladenosine (m^6^A) and its RMPs are summarized in Figure 2A; 5-methylcytosine (m^5^C) and its RMPs in Figure 2B; Adenosine to inosine (A-to-I) RNA editing and its RMPs in Figure 3A; Cytosine to uracil RNA editing and its RMPs in Figure 3B.

### 2.2. 5′Cap RNA Modifications and RMPs

In eukaryotes, a newly synthesized mRNA (pre-mRNA) must undergo three steps to become a mature mRNA, which include (1) capping at the 5′ end, (2) adding a poly(A) tail to the 3′ end and (3) splicing to remove introns [14]. The 5′ cap addition is a complex process, illustrated in Figure 4, concluding with a final m^7^G0pppN1 structure [15]. Loaded by RNA guanine-N7 methyltransferase (RGMT), 7-methylguanosine (m^7^G) protects the pre-mRNA from degradation and is recognized by 5′ cap-binding complex (CBC). CBC binds the 5′ cap and interacts with transcription machinery and the RNA splicing complex as well as translation initiation factors and the 40S ribosomal subunit to co-ordinate the functions of transcription, splicing, RNA export and translation [15,16].

2′-O-methylation (Nm) is one of the most abundant RNA modifications and can occur on any base in all RNA species [17]. In about half of all capped and polyadenylated RNA molecules, mRNA and small nuclear RNAs (snRNAs) with a 5′ cap structure are further modified by cap methyltransferases 1 and 2 (CMTR1 and CMTR2) to add Nm on the first and second transcribed nucleosides, respectively [18,19] (Figure 4). Cytoplasmic sensors MDA5 and RIG-I, which recognize non-2′-O-methylated viral RNAs, highlight an important function of Nm in the mRNA 5′ cap as a distinction between self and non-self [20,21,22,23].

N6-methyladenosine (m^6^A) is one of the most prevalent modifications in the 5′ cap of mRNAs [24], and the 5′ cap m^6^A stabilizes mRNA transcripts through resistance to the mRNA-decapping enzyme DCP2 [25]. The previously-known phosphorylated carboxyl-terminal domain (CTD) of RNA polymerase II (RNAP II) interacting factor (PCIF1) was recently found to be a 5′ cap-specific adenosine N6-methyltransferase (CAPAM), which catalyzes m^6^A of the first transcribed nucleoside adenosine to form the cap structure m7G(5′)ppp(5′)m6Am [26] (Figure 4). CAPAM/PCIF1 negatively regulates RNAPII-dependent transcription [27]. CAPAM/PCIF1 is the only known cap-specific adenosine N*6-*methyltransferase that does not methylate adenosine residues in the RNA body and its function is the subject of ongoing research [28].

### 2.3. Internal m^6^A and Associated RMPs

m^6^A is one of the most abundant modifications on mRNA and was first discovered in 1965 [29]. m^6^A exhibits tissue-specific distribution and is enriched near stop codons and in 3′ untranslated region (3’-UTRs) [30]. Found in all RNA types, including mRNA, tRNA, rRNA and non-coding RNA (ncRNA), m^6^A is involved in nearly every aspect of the mRNA life cycle and cellular processes, as well as cell/organ development and disease pathogenesis [11,13]. The m^6^A associated RMPs, i.e., writers, readers and erasers, are listed in Figure 2A. Methyltransferase 3 (METTL3) and methyltransferase 14 (METTL14) form a stable heterodimer that can efficiently catalyze methyl group transferring from *S*-adenosylmethionine (SAM) to the *N*^6^-amine of adenosine in the sequence motif *RRACH* (where R = G/A, H = A/C/U) of RNA [31,32,33]. The Wilms tumor-associated protein (WTAP) complex can interact with the METTL3/14 complex to promote mRNA methylation [31,34] (Figure 2A). Of note, recent studies showed that METTL16 binds to U6 snRNA and other ncRNAs as well as numerous lncRNAs and pre-mRNAs at a sequence motif, which is different from the METTL3/METTL14 binding motif [35,36], to regulate splicing and mouse embryonic development [35,37].

Readers of RNA m^6^A modifications primarily include members of the YTH family proteins and heterogeneous nuclear ribonucleoproteins (hnRNPs) (Figure 2A) [38,39,40,41]. YTH family proteins are mainly involved in translation regulation and degradation, either promoting m^6^A-modified RNA translation (YTHDF1 and 3, YTHDC2) or regulating m^6^A-modified RNA splicing and degradation (YTHDF2 and 3, YTHDC1) [38,42,43,44]. Similarly, hnRNPs are involved in regulating RNA processing/alternative splicing, and HNRNPC and HNRNPG facilitate m^6^A binding by targeting m^6^A-induced structural changes in RNA structure [40]. Additional m^6^A readers have been identified, such as IGF2BP1-2 and Prrc2a, which stabilize and promote storage of m^6^A-modified mRNA, further affecting gene expression [45]. Finally, fat mass and obesity associated (FTO) and ALKBH5 have been identified as m^6^A erasers [46,47,48]. FTO and ALKBH5 are known to regulate pre-mRNA alternative splicing and subsequent mRNA processing, metabolism, and export, respectively [48,49].

### 2.4. m^5^C and Associated RMPs

5-methylcytosine (m^5^C) has been well-characterized in tRNA and rRNA [50,51] and is now increasingly recognized as a widespread RNA modification in coding and non-coding eukaryotic mRNA as well [52,53,54]. However, m^5^C levels in mRNA remain low, reportedly ~0.43% by one early study [52]. The low abundance of m^5^C in mRNA is an obstacle in determining m^5^C function in transcription regulation [50], but the abundance of m^5^C methyltransferases and readers has revealed some insight into this concept. RNA m^5^C is catalyzed by RNA cytosine methyltransferases (RCMTs) (Figure 2B). This group is comprised of NOL1/NOP2/sun (NSUN) family members 1–7 and DNA methyltransferase 2 (DNMT2) [55]. RCMTs are known to play a wide variety of roles, notably ribosomal assembly, transcription regulation, mRNA nuclear export and tRNA cleavage and charging [55]. NSUN2 is known to be responsible for the majority of m^5^C in mRNA [56,57]. NSUN6 has been shown to be implicated in the formation of m^5^C in mRNA [58].

ALYREF has been shown to selectively bind m^5^C and is critical for exporting mRNAs that are methylated by NSUN2 [59]. Other proteins, which function as m^5^C readers, include YTHDF2, and Y-box binding protein 1 (YBX1) [60,61]. Together, these m^5^C readers are known to selectively bind m^5^C to regulate transportation, ribosomal biogenesis, and mRNA stability (Figure 2B).

Ten–eleven translocation (TET) enzymes are traditionally known to “reverse” DNA cytosine methylation by oxidizing m^5^Cs to 5-hydroxymethylcytosines (5hmCs). Recently TET1/2 enzymes have been shown to share partial responsibility for RNA m^5^C demethylation to 5-hydroxymethylcytosine (hm^5^C) in tRNA to promote translation [62] and to regulate embryonic stem cell (ESC) self-renewal network regulation [63].

### 2.5. RNA Editing: A-To-I and C-To-U

RNA editing is another common RNA modification. There are two common forms of RNA editing in higher eukaryotes, known as adenosine-to-inosine (A-to-I) and cytosine to uracil (C-to-U) editing [64]. Both A-to-I and C-to-U are important for codon alteration, as well as implicated in RNA splicing and mRNA processing, stability, localization and translation [64].

A-to-I editing is carried out by adenosine deaminase acting on RNA (ADAR) enzymes 1–3 (Figure 3A), and the vast majority of A-to-I sites are in noncoding sequences, 5′ and 3′ UTRs, and intronic retrotransposon elements, such as Alu [64]. Globally, ADAR1 is the primary editor of repetitive sites and ADAR2 is the primary editor of non-repetitive coding sites, whereas ADAR3 predominantly acts as an inhibitor of editing [65].

Occurring within regions of double-stranded RNA (dsRNA), A-to-I editing causes disruption and destabilization of dsRNA base pairing and affects the RNA interference pathway (RNAi), RNA splicing, and RNA secondary structures in a tissue-specific manner [66,67,68].

ADAR1-mediated A-to-I editing of endogenous dsRNA is needed for normal development [66] and essential in the prevention of cytosolic innate immune system activation [69]. ADAR3-mediated editing occurs primarily in the brain and is found to be overexpressed in glioma cells [70], as well as involved in mouse learning and memory [71]. Interestingly, A-to-I editing has been found to occur in a negative correlation with m^6^A [72].

C-to-U editing involves the hydrolytic deamination of a cytosine to a uracil base by a member of the AID/APOBEC enzyme family (Figure 3B). All members (apart from APOBEC2 and 4) demonstrate C-to-U deaminase activity on single-stranded DNA or RNA [73]. Genome-wide sequencing studies have shown that C-to-U editing is widespread, predominantly in 3′ UTR and coincident with splicing and polyA sites [74]. When in coding regions, such editing can create start or stop codons or alter amino acid codes and splice sites, leading to varied genetic information on the RNA level [75,76]. Finally, APOBEC1 and ACF regulate nonsense-mediated RNA decay [77]. Taken together, these studies suggest C-to-U editing functions in the regulation of mRNA processing, stability, localization and translation.

## 3. Transcription and Nascent RNA-Associated Chromatin Structure

### 3.1. Transcription and Nascent RNA Synthesis

Transcription is the first step of an essential multistep bioprocess and consists of three major steps: initiation, elongation and termination [78]. In eukaryotes, the activity between transcription initiation and elongation can be further described in five major steps: (1) chromatin remodeling; (2) formation of the preinitiation complex (PIC); (3) formation of an open complex and abortive synthesis of short (2–8) nt RNAs; (4) promoter pausing and escaping by RNAPII; and (5) pause releasing and productive elongation (Figure 5). In this section, we will describe these steps in detail.

Chromatin remodeling begins with the interaction between sequence-specific transcription factors (TFs) and chromatin modifiers to “open” condensed chromatin (heterochromatin) [79] (Figure 5, step 1), allowing RNAPII/Mediator complex to bind specific DNA sequences and form a PIC at the gene promoters and enhancers [79] (Figure 5, step 2). The mediator of RNAPII transcription (mediator) complex is critical in transcription activation and is assembled at the PIC. Interactions between RNAPII and mediator complex are facilitated by the C-terminal domain (CTD) of the largest unit, RPB1, of RNAPII, which contains a highly conserved region with up to 52 repeats of the heptad of Y_1_S_2_P_3_T_4_S_5_P_6_S_7_ (YSPTSPS) largely unphosphorylated during initiation [80] (Figure 5, step 2). Open complex formation is initiated by the multiprotein complex TFIIH. TFIIH can not only unwind DNA to form the transcription “bubble” through its associated helicases (Figure 5, step 3), but can also phosphorylate the YSPTSPS CTD serine 5 residue through its associated CDK7 and cyclin H kinases, stimulating mediator complex and PIC general TF dissociation and allowing RNAPII escape from the promoter [81,82] (Figure 5, step 3). Two other factors, DRB sensitivity inducing factor (DSIF) and negative elongation factor (NELF) contribute to the transition from transcriptional initiation to elongation and RNAPII promoter escape [83]. DSIF is a heterodimer composed of Spt4 and Spt5, and NELF consists of four subunits (A, B, C/D, and E) [84,85]. Spt5 interacts with the 5′ cap of nascent RNA (pre-mRNA), and then NELF recognizes the Spt5 interface [86]. TFIIH interacts with DSIF and NELF, which then cooperatively induce RNAPII promoter escape and subsequent pausing (Figure 5, step 4). Finally, transcription elongation is promoted by P-TEFb [87], which consists of a kinase/cyclin pair of CDK9 and CCNT1 (or CCNT2). P-TEFb phosphorylates DSIF and RNAPII CTD serine 2, inducing NELF release from RNAPII [88] and allowing RNAPII interaction with other elongation factors and chromatin modifying complexes to ensure productive transcription elongation, respectively [89,90,91] (Figure 5, step 5).

### 3.2. R-Loops as the Regulators of Transcription and Chromatin

#### 3.2.1. Mechanisms of R-Loop Formation

R-loops are three-strand nucleic acid structures consisting of an RNA-DNA duplex and an unpaired DNA strand generated during transcription upon nascent RNA “threadback” invasion into the DNA duplex to displace the non-template strand [92] (Figure 5). Different from the RNA: DNA hybrids occurring inside the active sites of RNAP during replication and transcription, R-loops span a much longer range (~100–200 base pairs) and form outside of RNAP during transcription [93]. R-loops are found ubiquitously from bacteria to mammals and frequently at gene promoters and enhancers, as well as terminators. Functionally, R-loops can positively and negatively regulate transcription and affect the differentiation and lineage plasticity of pluripotent progenitors/stem cells [94,95,96].

#### 3.2.2. Distribution and Function of R-Loops

R-loops prevent DNA methylation by DNMT3B1 and are found to be clustered at the CpG islands (CGIs) of gene promoters and enhancers in the human genome [97]. In transcription activation, R-loops are found to promote recruitment of epigenetic modifiers and RNAPII to CGIs [98]. As shown in Figure 5, R-loops at gene promoters and enhancers are associated with active chromatin markers, such as H3 lysine 27 acetylation (H3K27ac) and H3 lysine 4 trimethylation (H3K4me3), which suggests the involvement of R-loops in enhancer–promoter looping and high-order chromatin architecture [95,99].

Antisense RNA transcripts promote the formation of R-loops, creating a local chromatin environment which facilitates the recruitment of transcription factors and chromatin modifiers to promote gene transcription [100,101]. For example, the LncRNA TARID is reportedly involved in *TCF21* transcriptional activation by generating an R-loop at the CpG-rich promoter of *TCF21*, enabling GADD45A binding to the R-loop and recruitment of the DNA demethylase TET1 [102]. On the other hand, it is also well-documented that R-loops can trigger chromatin condensation and heterochromatin formation to repress gene expression [103]. Co-transcriptional R-loops have been shown to impede the movement of RNAP in vitro and exert a negative impact on transcription elongation [104]. R-loops were reported to regulate the chromatin structure of specific genes involved in control of mouse ESC differentiation by differentially interacting with the active Tip60-p400 histone acetyltransferase complex and repressive PRC2 H3K27me3 complex [95]. Mechanistically, higher Tip60-p400 and lower PRC2 levels are associated with promoter-proximal R-loops and active transcription to promote ESC differentiation, while decreased Tip60-p400 and increased PRC2 disrupts R-loop formation and impairs ESC differentiation [95].

In addition to gene promoters and enhancers, R-loops are highly enriched in retrotransposons (i.e., LINE-1), ribosomal RNA (rRNA), tRNA loci and mitochondria, and are crucial for regulating genome stability, stress response and mitochondrial function [105,106,107]. R-loops can cause hypermutation, hyperrecombination, chromosomal rearrangements or chromosome loss, leading to genome instability and chromosome fragility [108]. Genome-wide mapping of R-loops demonstrates an enrichment at the 3′ end of genes with GC skew patterns in mammalian cells, and R-loop accumulation at the 3′ end of genes inhibits transcription [109]. R-loops were found to be associated with a subset of developmental regulator genes, which are targeted by Polycomb complexes PCR1 and PCR2 [110]. Removal of the R-loop causes decreased PRC1 and PRC2 binding and increased RNAPII recruitment, resulting in activation of these PRC1/2 targeting genes [110]. Arginine methyltransferases PRMT1/5 and CARM1 interact with tudor domain-containing protein 3 (TDRD3) to facilitate recruitment of topoisomerase IIIB (TOP3B) to prevent such R-loop accumulation [111], and release RNAPII for successive transcription by methylating the arginine residues in histone tails, as well as the RNAPII CTD [112].

It is increasingly clear that the formation, distribution and removal of R-loops is tightly regulated during DNA transcription and replication.R-loops are involved in regulation of all stages of gene expression, from transcription initiation to termination, and can act as transcriptional activators or repressors at specific gene loci in different cell types and differentiation states. Regulatory R-loops are important for transcription elongation and termination, as well as telomere stability and DNA repair in normal physiological conditions. In contrast, dysregulated R-loops are associated with transcription elongation defects, DNA damage and genome instability in pathophysiological conditions, such as cancer [92,105].

### 3.3. MYC, 7SK snRNP and BRD4 Transcription and Chromatin Structure Regulation

#### 3.3.1. MYC-Mediated RNA 5’ Capping, Transcription Elongation and Active Chromatin

MYC is well-known to activate Myc-dependent transcription, but its underlying mechanisms remain incompletely understood [113,114]. As shown in Figure 6, MYC is involved in m^7^G 5′ capping of pre-mRNAs by binding to the E-box in the promoter of active genes, which recruits the capping enzyme complex composed of RNA guanylyltransferase and 5’-phosphatase (RNGTT) and CMTR1 [114]. Upon binding to the E-box, MYC also recruits CDK7/TFIIH and CDK9/P-TEFb to gene promoters, which phosphorylate RNAPII CTD serine 5 and 2 residues, respectively [115].

Finally, MYC appears to be implicated in regulation of R-loop formation as well. Recent studies have shown that MYC can suppress R-loop formation in promoter-proximal regions through recruitment of BRCA1, which stabilizes the DCP2 de-capping enzyme complex and enables RNAPII escape from promoter pausing, leading to increased transcription elongation and gene expression [116] (Figure 6).

#### 3.3.2. SK snRNP Complex as a Negative Regulator of Transcription Elongation

The 7SK small nuclear ribonucleoprotein (7SK snRNP) complex is composed of the core 7SK snRNPs and other proteins, including AFF1/4, HEXIM1/HEXIM2, LARP7, MePCE and hnRNPs [117]. This complex functions as a major negative regulator of CDK9/P-TEFb by inhibiting CDK9/P-TEFb-mediated RNAPII escape and transcription elongation [118,119]. 7SK snRNP interacts with HEXIMI that directly binds CDK9/P-TEFb, inactivating its kinase activity and inhibiting transcription elongation [120] (Figure 6).

#### 3.3.3. BRD4-Mediated Transcription Elongation and Active Chromatin

BRD4 is a member of the Bromodomains and Extraterminal (BET) family and contains two bromodomains that recognize acetylated lysine residues of histone tails [121,122]. BRD4 not only promotes the establishment of chromatin, favoring transcriptional activation, but also directly competes with release of CDK9/P-TEFb sequestrated by the 7SK snRNP complex, which promotes CTD-S2 phosphorylation and activates RNAPII-dependent transcription elongation [123] (Figure 6). BRD4 was initially reported to interact with CDK9/P-TEFb through its bromodomain, resulting in CDK9/P-TEFb-dependent phosphorylation of RNAPII CTD serine 2 (CTD-S2) and transcription activation in vivo [124,125]. However, later studies demonstrated that BRD4 functions as an atypical kinase by directly binding and phosphorylating CTD-S2 both in vitro and in vivo under conditions where other CTD kinases are inactive [126]. Our study demonstrated that RNA cytosine methyltransferase NOP2/NSUN1 and NSUN2 interact with BRD4 and the elongating form of RNAP (CTD-S2P) to form a transcriptionally active chromatin structure in leukemia cells [127] (Figure 6). BRD4 was also found to prevent R-loop accumulation and to protect against transcription-replication collision [128]. Loss of BRD4 in cancer cells leads to replication stress, DNA damage and eventually apoptotic cell death [128].

## 4. Co-Transcriptional RNA Splicing and Its Associated Chromatin Structure

### 4.1. RNA 5′ Capping Enzyme-Coupled Spliceosome Assembly

Over 90% of human genes undergo alternative splicing, which is carried out by splicing machinery called the spliceosome. The spliceosome contains more than 150 associated proteins and five uridine-rich small nuclear RNAs (snRNAs)—U1, U2, U4, U5 and U6 (RNU1, RNU2, RNU4, RNU5 and RNU6) [129,130]. Nascent RNA splicing is regulated by multiple *cis*-elements and *trans*-factors through extensive interactions between splicing factors and the core transcription machinery [131,132,133,134,135,136]. At the earliest stages of spliceosome assembly (Figure 7A), the CTD serine 5 phosphorylation (S5P) of RNAPII facilitates recruitment of the 5′ capping enzyme complex (CE/RGMT). The U1 snRNP binds to the GU sequence at the 5′ splice site (5′SS) of the first intron of mRNA through base pairing, and splicing factors U2AF1 and U2AF2 bind at the 3′SS and the branch point sequence (BPS), respectively [137]. The U2 snRNP complex interacts with the BPS and the pre-mRNA 3′SS site of pre-mRNA, and the SF3A and SF3B proteins in the U2 snRNPP complex stabilize these interactions [138]. The U5/U4/U6 snRNP trimer is recruited with the U5 binding to the 5’SS and the U6 binding to U2 (Figure 7A). Then the U1 snRNP is released, U5 shifts from exon to intron, and the U6 binds at the 5’ splice site (Figure 7b). The splicing factor SF2/ASF binds the S2-phosphorylated CTD of RNAPII to facilitate the interactions between the spliceosomal U1 and U2 snRNPs complexes as well as RNAPII during the splicing process [139,140].

### 4.2. Alternative Spliceosome Assembly and Determination of Internal Exon/Intron Junctions

Determination of the internal exons and introns of pre-mRNAs, i.e., alternative splicing, is carried out by the spliceosome through extensive interactions between *trans*-acting factors/complexes and *cis*-acting spicing elements/sequences, as well as transcription machinery and chromatin modifying complexes [132,135,141,142,143,144,145,146,147]. Serine and arginine rich splicing factors (SRSFs) selectively bind to *cis* exonic and intronic splicing elements, promoting spliceosome formation [148], while hnRNPs, such as hnRNPA1 (HNRNPA), negatively regulate spliceosome formation by binding to exonic or intronic splicing silencers [149] (Figure 7B).

### 4.3. Alternative RNA Splicing Regulation by Transcription Elongation Complex

Transcription elongation rate and chromatin structure have a large impact on splice site identification and definition of exons and introns by the spliceosome [143]. Recent studies show depletion of the pre-exon junction complex (EJC), a protein complex formed at the junction of two exons of a pre-messenger RNA during RNA splicing, led to a global decrease in RNAPII pausing and premature entry into elongation by CDK9/P-TEFb, suggesting that the pre-EJC serves as an early transcriptional checkpoint to prevent premature entry into elongation and proper exon definition [150].

Other *trans-*acting factors, such as MYC, BRD4 and many RNA-binding proteins (RBPs) regulate transcription elongation and alternative splicing in tissue-specific patterns and sizes [151,152]. MYC and BRD4 have been shown to physically interact with RNA splicing factors to contribute to spliceosome formation and alternative splicing regulation at the co- and post-transcriptional levels [153,154,155,156,157], as shown in Figure 7.

### 4.4. Nascent Cis-Acting Elements and Degradation Machinery Regulate Alternative RNA Splicing

Intronic RNAs constitute most pre-mRNAs in vertebrate cells and were traditionally considered transcriptional “junk” or “waste” due to their complete degradation first by a unique debranching endonuclease, followed by exoribonucleases after splicing [158]. However, recent studies show that many spliced introns can escape complete degradation and are further processed to generate regulatory RNAs that can regulate chromatin structure, gene expression, and genomic stability [159] (Figure 7).

Terminal exons are defined by the cleavage and polyadenylation complex (CPA), which is composed of the cleavage and polyadenylation specificity factor (CPSF) complex, cleavage factors (CFs) and the cleavage stimulation factor (CstF) complex. The CPSF complex recognizes the polyadenylation signal sequence AAUAAA and other nearby sequences [160] (Figure 7B). The poly(A) tail is added to nascent RNA when the nascent RNA is cleaved to produce mature RNA by the polyadenylate polymerase complex, which is also physically linked to the spliceosome [161]. After introns are spliced out from pre-mRNA, exons are ligated together to yield multiple mature mRNAs through alternative splicing, significantly increasing transcriptomic/proteomic complexity [162].

### 4.5. The Impact of Chromatin Structure and R-Loops on Alternative RNA Splicing

Earlier studies showed that nucleosome positioning contributes to definition of exons and introns, and the ratio between nucleosome occupancy within and upstream from the exons correlates with exon-inclusion levels [163,164,165]. Recent studies have demonstrated that splicing machinery interacts with histone and DNA modifications and their modifiers, such as histone acetyltransferases (HAT), histone deacetylases (HDACs), histone methyltransferases (HMTs), DNA methylation, and DNA methyltransferases (DNMTs) [166,167] (Figure 7A,B). Using HDAC inhibitors (HDACIs), such as trichostatin A (TSA) or knockdown of HDAC1 or HDAC2 increased H4 acetylation, specifically at the exon 2 of *MCL1*, a gene encoding a longer form of antiapoptotic protein (MCL1_L_) and a shorter form of proapoptotic protein (Mcl-1_S_), resulting in exon 2 skipping and production of the short form of proapoptotic MCL1_S_ [168,169]. Altered histone methylations and methyltransferases are associated with differential exon inclusion and intron retention [170,171]. ZMYND11, aka BS69, is a reader for histone H3.3 lysine 36 trimethylation (H3K36me3) and involved in transcription elongation. It is known to interact with U5 snRNP in the spliceosome, and depletion of BS69/ZMYND11 strongly decreases intron retention [172].

Other studies have shown that DNA methylation and DNA modifiers are also involved in RNA splicing [173]. Higher levels of DNA methylation have been identified in exons, and especially splice sites, than those found in flanking introns [174]. The zinc-finger protein CCCTC-binding factor (CTCF) negatively regulates RNAPII elongation rate through transient obstruction of RNAPII and promotes splicing by favoring spliceosome assembly at weak splice sites. DNA 5-methylcytosine (5mC) promotes exon exclusion through 5mC-mediated CTCF eviction; however, TET1 and TET2-mediated 5-hydroxymethylcytosine (5hmC) has an opposite effect on CTCF-dependent splicing [175] (Figure 7). The methyl-CpG binding protein 2 (MeCP2) was shown to interact with the RNA-binding protein Y box-binding protein 1 (YBX1) and to regulate alternative splicing in Rett syndrome (RTT) mouse models [176]. Additionally, a recent study demonstrated that R-loops are able to activate gene promoters by recruiting the DNA methylcytosine dioxygenase TET1 to demethylate CpG islands [102].

## 5. Chromatin Associated Non-Coding RNAs and Acting Modes in Chromatin Remodeling

### 5.1. Non-Coding RNAs (ncRNAs) Subtypes

The advent of genome-wide sequencing technologies has revealed that 75% of the human genome is transcribed into RNA, but only a small percentage of transcribed RNA encodes proteins [177,178]. Noncoding RNAs play an important role in regulating gene expression and chromatin structure and can be functionally or structurally divided into several subtypes [179,180,181,182,183,184], as shown in Table 1.

As stated previously, ncRNAs were traditionally considered transcriptional “junk” or “waste”, but are now widely accepted to play an important role in regulation of transcription elongation and chromatin structure [185]. It is well-known that ncRNAs can be generated from gene bodies, extragenic regions (particularly enhancers) [186,187], and promoter upstream sequences to generate a class of short, polyadenylated, and highly unstable RNAs called promoter upstream transcripts (PROMPTs) [188,189]. Expression levels of enhancer RNAs (eRNAs) are correlated with signal-regulated transcriptional activity of gene enhancers. Furthermore, the heterogeneity of eRNAs may also be correlated with their functional diversity in the mammalian genome [187,190].

### 5.2. Non-Coding RNAs Regulate Chromatin Structure and Transcription in Cis

The *cis-*acting models of ncRNAs include formation of R-loops and DNA-RNA triple helixes through Hoogsteen base-pairing interactions between the major groove of double-stranded DNA and single stranded RNA [191,192,193]. Particularly, the *cis*-acting symmetrical triplex-forming motifs in lncRNAs favor the formation of RNA-DNA triplexes, which can affect RNA-RNA, RNA-protein, and RNA-DNA interactions and modulate chromatin structure [191]. By forming RNA: DNA: DNA triplexes/R-loops, lncRNAs can target specific DNA sites, such as enhancers, to contribute to local chromatin organization [194,195].

### 5.3. Non-Coding RNAs Regulate Chromatin Structure and Transcription in Trans

The most common mode of ncRNA-mediated chromatin organization is through *in trans* RNA-protein interactions. The role of RBPs in regulating chromatin structure and gene expression has been studied extensively [196,197]. RBPs may have single or multiple previously characterized canonical RNA binding domains (RBDs), including zinc finger domains [198,199], RNA recognition motif (RRM) [200], double-strand RNA binding domain (dsRBD) [201] and hnRNPK homology domain (aka KH) [202,203]. As one of the best-characterized RBPs, hnRNPK is known to play a role in regulating transcription and splicing, as well as chromatin structure and DNA damage and repair [204,205,206,207]. Other RBPs are known to be involved in chromatin remodeling through interaction with DNA methyltransferases (DNMTs) or methyl-DNA-binding domain (MBD)-containing proteins, such as Methyl-CpG binding protein 2 (MeCP2) [208,209]. MeCP2 is a multifunctional epigenetic reader that can bind RNA and methylated CpGs in DNA to orchestrate chromatin compartmentalization and higher order genome architecture [209].

Recent genome-wide proteomic studies have identified many new RBPs [2,210]. Those newly identified RBPs are conserved from yeast to humans. Despite lacking the conventional or canonical RBDs, those new RBPs can modulate chromatin structure and gene expression in response to various environmental and intrinsic stimuli [2,210,211,212,213]. Both canonical and non-canonical RBPs are reported to function together to regulate transcription and chromatin structure [214,215].

## 6. Fine-Tuning Chromatin and Transcription through RNA Modifications and RNA-Modifying Proteins

### 6.1. m^6^A and RMPs Modulate Interactions between Non-Coding RNAs, Transcription Factors and Chromatin Modifiers

The METTL3/14 methyltransferases, YTH-domain-containing proteins (YTHDs)/HNRNPs, and FTO/ALKB family proteins function as m^6^A writers, readers and erasers, respectively, to regulate the level and distribution of m^6^A in a tissue-/differentiation-specific manner [31,34,38,46,216] (Figure 2A). Recent studies have shown widespread m^6^A in human tissues, which is significantly enriched in long intergenic non-coding RNA (lincRNA) and CpG-rich gene promoters, suggesting that m^6^A is positively correlated with gene expression homeostasis and has broad involvement in human development and disease [217].

Increasing evidence suggests that m^6^A and its associated RMPs, more specifically m^6^A writers and readers, modulate chromatin structure and transcriptional activation through interactions with TFs and chromatin modifying proteins/complexes [218,219]. METTL14 was reported to recognize the active histone mark H3K36me3, resulting in METTL3-METTL14-WTAP m^6^A methyltransferase complex recruitment to the vicinity of H3K36me3 peaks, promoting m^6^A and positively regulating transcription elongation [220] (Figure 8A).

Zinc finger protein 217 (ZFP217), a chromatin-associated oncogenic TF important for embryonic stem cell (ESC) differentiation [221], is associated with repressive chromatin marked by histone 3 lysine 9 trimethylation (H3K9me3) and histone 3 lysine 27 trimethylation (H3K27me3) through interactions with G9a and EZH2 methyltransferases [8]. ZFP217 also negatively affects m^6^A deposition at its own target gene mRNAs by sequestering METTL3 in an inactive status, in which METTL3 is dissociated from the METTL14/WTAP co-factor complex [221,222] (Figure 8A).

YTHDC1 can de-repress (activate) gene expression by physically interacting with and recruiting KDM3B demethylase to hyper-m^6^A in nascent RNAs, leading to demethylation of the repressive transcriptional histone modification H3K9me2 [223]. YTHDC1 can also facilitate the decay of a subset of these m^6^A-modified RNAs, especially LINE-1, through nuclear exosome targeting-mediated nuclear degradation, while METTL3 deposits m^6^A modifications on chromosome-associated regulatory RNAs (carRNAs), including promoter-associated RNAs, enhancer RNAs and repeat RNAs [224].

m^6^A was reported to promote translation of the SETD1A/B components, which increases the transcriptional activation histone mark H3K4me3 and recruitment of the erythroid-specific zinc finger transcription factor KLF1 to its target gene promoters [225]. Inhibition of the METTL3/METTL14/WTAP m^6^A methyltransferase complex blocks erythropoiesis in human bone marrow hematopoietic stem/progenitor cells (HSPCs). The METTL3-METTL14-WTAP complex also functionally interacts with SMAD2/3 proteins, the key signal transducers and transcriptional modulators of the TGFβ pathway. SMAD2/3 promotes binding of the m^6^A methyltransferase complex to a subset of transcripts involved in early cell fate decisions [226]. In addition, a recent study showed that RBM15, a subunit of the m^6^A methyltransferase complex, mediates the degradation of the pre-BAF155 mRNA to affect the chromatin remodeling function of the BAF (SWI/SNF-like) complex [227].

### 6.2. The Impact of m^5^C and Its RMPs on Chromatin and Transcription

In eukaryotes, RNA m^5^C is catalyzed by RCMTs, including NOL1/NOP2/SUN domain (NSUN) family enzymes and DNMT2 (Figure 2B). NSUN2 is responsible for catalyzing the majority of m^5^C in mRNAs and non-coding RNAs [57,228,229]. Our study demonstrated a marked increase in NOP2/NSUN1 and NSUN2-mediated m^5^C in drug (5-azacitidine) resistant myeloid leukemia cells [127]. NSUN7 is reported to promote transcription of genes labeled with enriched m^5^C eRNAs in mouse hepatoma cells and primary hepatocytes for adaptive metabolic responses [226]. Mechanistically, the methyltransferase SET7/9 methylates histone 3 K4 residues [227] and PGC-1α K779, a transcriptional co-activator important for regulating adapted metabolic responses [230]. Methylated PGC-1α interacts with Spt-Ada-Gcn5-acetyltransferase (SAGA) and mediator complexes, as well as NSUN7, at the m^5^C-marked enhancer RNAs of PGC-1α target genes to promote recruitment of transcription machinery and reinforce transcription of those genes (Figure 8).

Heterozygous mutations in the X-linked gene encoding MeCP2 cause the neurological disorder Rett syndrome. MeCP2 interacts with more than 40 binding partners, including transcriptional regulators, chromatin modifiers and RNA- splicing factors [231]. MeCP2 chromatin binding is controlled by m^5^C, and recent studies show enrichment of MeCP2 on long non-coding RNAs (lncRNAs), such as retinal noncoding RNA3 (RNCR3) [232], muscle-specific long non-coding RNA (ChRO1) [233], and major Satellite Forward RNAs [234]. This MeCP2 enrichment on lncRNAs promotes deposition of H3K9me3 and H4K20me3 and heterochromatin formation. Additionally, m^5^C is present in a wide variety of RNA species, including cytoplasmic and mitochondrial ribosomal RNAs (rRNAs) and tRNAs, as well as mRNAs, eRNAs and other types of non-coding RNAs [55]. There are widespread and dynamic changes in RNA m^5^C in response to stress and stimuli, which suggests an important role of m^5^C and its RMPs in epigenetic gene regulation [55]. However, the detailed mechanisms underlying m^5^C and its RMP-mediated chromatin remodeling and gene regulation are yet to be elucidated.

### 6.3. RNA Modifications and RMPs in R-Loop Formation and Transcription Regulation

R-loops can be formed both *in cis* by a nascent RNA folding back to its DNA template at the same transcription locus and *in trans* by ncRNAs at distant enhancers (Figure 5). R-loops are prevalent in highly transcribed genes, and occur most frequently at conserved genetic hotspots, such as unmethylated CpG island promoters, G-rich terminators and other regulatory chromosomal loci [98,104]. R-loops play an important role in organizing chromatin structure and maintaining genome stability [99], and RNA modifications and their associated RMPs can modulate the interactions between chromatin-associated RNAs and chromatin modifiers, as well as transcription machinery, to regulate chromatin structure and transcription in a position and context-dependent manner [235].

#### 6.3.1. m^6^A and Its RMPs Promote R-Loop Formation at Transcription Termination Sites

A recent study showed that R-loops accumulate in m^6^A-rich transcription termination sites (TTSs) [236]. Depletion of m^6^A methyltransferase complex components dramatically reduces R-loop accumulation in m^6^A marked genes around TTSs, leading to disruption of proper transcription termination [236]. In contrast, depletion of the nuclear m^6^A reader protein YTHDC1 does not affect R-loop levels [236].

#### 6.3.2. m^6^A and RMPs Associated R-Loop Formation on DNA Damage-Associated RNAs

ATM serine/threonine kinase (ATM) is recruited to DNA double-strand break (DSB) sites to phosphorylate several key proteins, leading to cell cycle arrest, DNA repair or apoptosis [237]. Specifically, in response to DSBs, ATM phosphorylates METTL3 serine 43, and the phosphorylated METTL3 is then recruited to DNA-damage sites to catalyze m6A in the DNA damage-associated RNAs. This increased m^6^A deposition leads to YTHDC1 recruitment and R-loop accumulation, promoting homologous recombination-mediated DNA repair [238]. In human pluripotent stem cells, m^6^A-containing R-loops are reported to accumulate during G_2_/M and are depleted at G_0_/G_1_ phases of the cell cycle [239]. Knockout of the m^6^A reader YTHDF2 leads to increased R-loop levels, cell growth retardation, and accumulation of the DNA double-strand break marker γH2AX in mammalian cells, which suggests that YTHDF2 negatively regulates m^6^A and R-loop accumulation [239]. Taken together, these results suggest that m^6^A and its RMPs regulate accumulation of R-loops and genomic stability in a position and context-dependent manner [96,240,241].

#### 6.3.3. RNA Cytosine Methylations and RMPs in R-Loop Formation

R-loops have been found to be required for transcription-coupled homologous recombination, as they recruit Cockayne Syndrome Protein B (CSB) and RAD52 proteins, but not the canonical HR proteins BRCA1 and BRCA2, to sites of reactive oxygen species (ROS)-induced DNA damage [242]. The formation of R-loops at break sites is dependent upon Drosha, an RNase III enzyme in the microRNA (miRNA) biogenesis apparatus [243]. DNMT2 is predominantly located in the cytoplasm and mitochondria, where it contributes to m^5^C in tRNAs, consequently affecting protein translation [244,245,246]. Little is known about the role of DNMT2 in regulating chromatin structure and R-loop formation. A recent report showed that DNMT2 (Figure 3B) is recruited to DNA damage sites, where it deposits m^5^C onto the DNA damage-associated mRNAs and contributes to R-loop formation. The m^5^C and DNMT2-mediated R-loops facilitate the homologous recombination (HR) and DNA repair of DSBs [247].

Loss of TRDMT1 in cancer cells confers sensitivity to PARP inhibitors in vitro and in vivo, suggesting that transcription-coupled RNA modifications may serve as DNA damage markers to regulate DNA repair [247]. METTL8 was previously reported to contribute m^3^C in mRNA, while METTL2 and METTL6 contribute m^3^C in tRNAs [248]. Recently, METTL8 was found to form a large SUMOylated nuclear RNA-binding protein complex that is associated with R-loops, and knockout of METTL8 resulted in reduction of m^3^C and R-loops in the nucleolus [249]. The nucleolar-enriched SUMOylated METTL8 complex regulates R-loop formation and promotes tumorigenesis [249]. Retention of R-loops at their transcription sites of nascent RNAs is an important and intrinsic part of chromatin structure in human cells, and R-loop dysregulation is associated with DNA damage, transcription elongation defects, hyper-recombination and genome instability [105,250].

### 6.4. RNA Editing in Chromatin Remodeling

#### 6.4.1. A-To-I Editing and ADAR-Mediated Heterochromatin and Gene Silencing

A-to-I editing is one of the most abundant RNA modifications in mammalian cells and is catalyzed by ADAR family proteins ADAR1 and ADAR2 (Figure 3A). A-to-I editing occurs most frequently in non-coding RNAs within repetitive elements in the genome, mainly Alu repeats [251,252,253], and has a critical role in immune regulation [69,254], and alternative splicing [67]. Vigilin is the largest protein in the KH domain-containing RNA-binding protein family [255]. Vigilin binds to promiscuously A-to-I-edited RNAs, predominantly in inverted Alu repeats, and is involved in the formation of heterochromatin [256], which is marked with the repressive chromatin marker H3K9me3 by histone H3-specific methyltransferase SUV39H1 [257,258]. Mechanistically, vigilin binds to A-to-I-edited RNAs in a complex that contains ADAR1, as well as RNA helicase A and Ku86/70 [259]. The C-terminal domain of human vigilin interacts with SUV39H1 and recruits it to the A-to-I-edited RNAs, leading to formation of heterochromatin and gene silencing [259]. This suggests a mechanistic link between DNA-repair machinery and the A-to-I RNA editing complex. Interestingly, the levels of global A-to-I editing and m^6^A in mammalian cells were shown to be negatively correlated, and depletion of m^6^A modification increases the association of m^6^A-depleted transcripts with ADAR enzymes, resulting in upregulated A-to-I editing on the same m^6^A-depleted transcripts [72,260].

#### 6.4.2. C-To-U Editing and Associated RMPs in Chromatin Remodeling and Gene Regulation

C-to-U editing is carried out by the APOBECs family proteins (Figure 3B). APOBECs function to restrict retroviruses and retrotransposons [261], and have been identified as a prominent source of mutations in cancers [262]. APOBEC3B (A3B) was shown to regulate expression of the estrogen receptor α (ERα)-target gene by causing C-to-U deamination at ER binding regions [263]. Further, transient cytidine deamination by A3B promotes chromatin modification and remodeling at regulatory regions of ER target genes that increase ER expression levels, and elevated A3B expression is associated with poor patient survival in ER+ breast cancer [263]. However, more studies are needed to understand the role of C-to-U editing and the APOBECs in regulation of chromatin structure and gene expression.

## 7. Implications of RNA Epigenetics in Human Diseases

The advent of new genome-wide sequencing technologies has uncovered abnormal RNA modifications and their associated RMPs in a variety of human diseases. Here we will briefly summarize the recent progress surrounding the roles of some major RNA modifications and their associated RMPs in four major disease categories: neuronal disorders, metabolic disorders, immune disorders/infections and cancer/leukemia (Table 2).

### 7.1. RNA Epigenetics in Neural Disorders

#### 7.1.1. m^6^A and Its RMPs

m^6^A is highly enriched in neural tissue and plays an important role in neurogenesis and neuronal disorders [264,265,266,267] (Table 1). Knockout of *METTL3* or *METTL14* prolongs the cell cycle of embryonic neural stem cells (NSCs)/radial glia cells (RGCs) and inhibits proliferation of NSCs/RGCs, resulting in delay of cortical neurogenesis into postnatal stages in embryonic mouse brains [268,269]. The effects of m^6^A and its writers on NSC self-renewal and neurogenesis have been linked to histone modifications [270,271]. METTL3/METTL14 and the reader protein YTHDF1 are also important for maintaining the stemness of adult neural stem cells (aESCs) and axon regeneration in adult mice [272]. The m^6^A eraser protein FTO also plays a critical role in regulation of the proliferation and differentiation of adult neural stem cells (aESCs) [273,274].

Recent studies show that m^6^A and its RMPs, specifically METTL3, YTHDF1 and FTO, play an important role in regulation of stress responses, formation of hippocampal memories, and development of neuropsychiatric diseases [275,276,277]. Furthermore, the FTO-induced hypo-m^6^A levels have been shown to activate the TSC1-mTOR-Tau signaling pathway and increased apoptosis of dopaminergic neurons, which contributes to the pathogenesis of Alzheimer’s disease (AD) and Parkinson’s disease (PD), respectively [278,279,280].

#### 7.1.2. RNA Editing and Its RMPs

RNA editing can alter codons and/or splicing sites in pre-mRNAs, resulting in non-functional transcripts or proteins. Earlier studies demonstrated that knockout (KO) of ADAR2 expression could substantially reduce the functional mRNA transcripts from the mutated AMPA receptor gene and rescue mice from mutations-induced lethality [281,282]. More recent studies demonstrate that A-to-I and C-to-U RNA editing enzymes can affect the development and function of neurons and neurotransmitters through modulating mRNA abundance and miRNA binding of neuronal development-related genes and neurotransmitters in neurons [283,284,285] (Table 2).

#### 7.1.3. RNA 2’-O-Methylation (Nm) and Its RMPs

Nm is one of the most abundant RNA modifications [17]. Recently, the Nm writer proteins FTSJ1 and CMTR1 have been shown to be involved in intellectual disability and neural development through methylation in tRNA and the 5’ cap of pre-mRNA in neurons, respectively [286,287,288,289] (Table 2).

#### 7.1.4. m^5^C and Its RMPs

Eukaryotic m^5^C and its RMPs regulate the function of three RNA species: rRNA, tRNA and mRNA [55]. m^5^C and its writer proteins, NSUNs, are involved in early mouse embryogenesis, especially neurogenesis [290], and the embryonic NSCs of mouse brains shows a distinct m^5^C pattern in mRNA [53]. NSUN2, a major m^5^C writer protein, is highly expressed in early neuroepithelial progenitors of the developing human brain, and loss of NSUN2-mediated tRNA:m^5^C impairs differentiation and motility of NES cells in mouse models [291]. Mutations in the *NSUN2* gene cause aberrant tRNA:m^5^C and autosomal-recessive intellectual disability, called Dubowitz syndrome [292,293,294,295,296] (Table 2).

### 7.2. RNA Epigenetics in Metabolic Disorders

#### m^6^A and Its RMPs

Recent studies show that m^6^A and its RMPs play a critical role in Type 2 Diabetes (T2D) pathogenesis, and ablation of m^6^A levels by targeting METTL3 or METTL14 in mouse models recapitulates human T2D by inhibiting the Insulin/IGF1-AKT-PDX1 pathway in pancreatic β-cells, resulting in cell-cycle arrest and impaired glucose-stimulated insulin secretion *(*GSIS) [297]. METTL14 deletion in β-cells of adult mice results in glucose intolerance [298,299]. WTAP, coupled with METTL3 and METTL14, is increased and distributed in the nucleus through upregulation of cyclin A2, which positively regulates mitotic clonal expansion of adipocytes [300]. The zinc finger transcription factor ZFP217 promotes mitotic clonal expansion (MCE) of adipocytes and adipogenesis by inhibiting CCND1 mRNA expression, which is positively and negatively regulated by METTL3 and YTHDF2, respectively [301] (Table 2).

FTO is well-known to promote adipogenesis, and inactivation of the gene encoding FTO protects mice from obesity [302]. The underlying mechanism has been linked to *IRX3*, a member of the Iroquois homeobox protein family [303]. FTO-induced adipogenesis is achieved by MCE regulation via the adipogenic isoform of RUNX1T1 [304]. Other molecular mechanisms by which FTO promotes adipogenesis are inhibition of the Wnt/β-catenin signaling in intramuscular preadipocytes [305] and activation of ATG5 and ATG77-mediated autophagy and adipogenesis [306]. Finally, since FTO negatively regulates m^6^A levels and positively regulates adipogenesis, FTO inhibitors may provide a novel therapeutic strategy for metabolic diseases [307].

### 7.3. RNA Epigenetics in Immune Disorders and Viral Infections

#### 7.3.1. m^6^A and Its RMPs

Recent studies demonstrate a critical role of m^6^A in the regulation of immune disorders and infections [308,309]. The innate immune system detects the nucleic acids, i.e., DNA and RNAs, of various pathogenic organisms via pattern-recognition receptors (PRRs). PRRs include several endosomal members of the Toll-like receptor (TLR) family and cytosolic sensors, such as retinoic acid inducible gene-I (RIG-I) and its homolog melanoma differentiation associated gene 5 (MDA5), as well as RIG-I-like receptors (RLR) [310,311]. RNA modifications, such as m^5^C, m^6^A, m^5^U, s^2^U or pseudouridine (Ψ), are known to ablate activation of dendritic cells (DCs) to inhibit innate immune system through TLRs [312]. Specifically, METTL3-mediated mRNA:m^6^A was recently shown to promote dendritic cell activation [313]. METTL3 has also been shown to be involved in T cell homeostasis and differentiation [314]. Depletion of METTL3 in naive T cells decreased m^6^A in the mRNAs of SOCS family genes encoding the STAT signaling inhibitory proteins SOCS1, SOCS3 and CISH, leading to slower mRNA decay and increased protein expression, consequently inhibiting IL-7-mediated STAT5 activation in the Mettl3-deficient T cells [314]. Another study showed that deletion of METTL3/m^6^A leads to suppression of IL-2-STAT5 signaling pathway, which controls Treg cell functions and regulates the tumor-killing functions of CD8 T cells [315]. Yet further studies show that METTL3 or YTHDF2 deletion leads to an increase in type I interferon β (IFNB1) mRNA, thus suppressing the propagation of different viruses [316].

Depletion of METTL14 showed a similar effect, stimulating accumulation of dsDNA- or virus-induced IFNB1 mRNA and inhibition of virus reproduction. Alternatively, ALKBH5 had the opposite effect [317]. Since the viral Rev protein preferentially interacts with methylated RRE, decreased or increased m^6^A levels in HIV-1 RRE due to silencing of METTL3/METTL14 or ALKBH5, respectively, lead to reduced or increased viral RNA export and replication suppression [318] (Table 2).

#### 7.3.2. RNA Editing and Its RMPs

Most A-to-I editing is predominantly located within mobile elements in non-coding parts of the human genome [65] and plays a critical role in innate immunity and protection against self-transcript activation [69,319]. ADAR1 knockout in neuronal progenitor cells increases MDA5 (dsRNA sensor)-dependent spontaneous interferon production, PKR activation, and cell death [320]. Therefore, human ADAR1 regulates sensing of self vs non-self RNA, enabling pathogen detection and autoimmune/inflammation prevention. Loss-of-function mutations in the ADAR1 gene cause Aicardi-Goutières syndrome, a congenital autoimmune disease characterized by encephalopathy and IFN signature [321] (Table 2).

#### 7.3.3. 5′ Cap Nm and CMTR

The 5′ cap structure with m^7^G (cap 0) and Nm of the first residue (cap 1) is vital for pre-mRNA processing and translation in multicellular eukaryotes and some viruses (Figure 4). The human coronavirus mutants lacking 2’-O-methyltransferase activity induce IFN expression in an MDA5–dependent manner, and these mutants are more sensitive to the antiviral action of IFN [21,322]. IFIT1 (IFN-induced protein with tetratricopeptide repeats-1) is an important effector of innate immune and antiviral response. Its antiviral activity relies on its ability to selectively bind the viral mRNA 5’ cap that lacks Nm in order to compete with translation initiation factor eIF4F binding to the same mRNAs, while remaining inactive against host mRNAs with Nm at the cap 1 and 2 residues [323,324,325,326] (Figure 4). A recent study showed that the SARS-CoV-2 nsp16 and nsp10 complex can carry out the 5′ cap Nm on virally encoded mRNAs to mimic cellular mRNAs, thus protecting the virus from host innate immune restriction [327]. This may provide a potential strategy to develop coronavirus vaccines and antiviral drugs [328].

### 7.4. RNA Epigenetics in Cancer and Leukemia

#### 7.4.1. m^6^A and Its RMPs in Cancer

Recent studies have demonstrated that RNA modifications and RMPs play a critical role in controlling cancer cell proliferation, metastasis, immune evasion and drug resistance [329,330,331]. m^6^A and its RMPs have a bi-faceted role, i.e., acting as either an oncogene or a tumor suppressor in the pathogenesis of solid tumors and leukemia. METTL3 upregulation increases translation of MYC, BCL2 and PTEN mRNAs in acute myeloid leukemia (AML), and METTL3 loss leads to increased levels of phosphorylated AKT, promoting differentiation of hematopoietic stem/progenitor cells (HSPCs) [332] (Table 2). METTL3, independently of METTL14, interacts with the transcription factor CEBPZ and binds the gene promoters occupied by CEBPZ to promote m^6^A in the CEBPZ-targeting gene transcript, enhancing translation and thereby promoting leukemogenesis [333].

m^6^A and its RMPs are essential for mRNA processing and miRNA biogenesis [334]. Upregulation of METTTL3 and m^6^A has been reported to promote tumorigenesis or metastasis through promotion or inhibition of oncogenic or tumor suppressive miRNAs biogenesis, respectively, in various solid tumors, including bladder [335], lung [336], breast [337], gastrointestinal and liver cancers [338,339], and glioblastoma [340]. However, elevated METTL3 expression inhibits self-renewal of bladder tumor initiating cells and bladder tumorigenesis through Notch 1 mRNA:m^6^A [341].

In contrast to increased m^6^A/METTL3 in most tumor types, decreased METTL14/m^6^A has been reported to act as a tumor suppressor, and elevated METTL14 expression has been reported in a variety of solid tumors and leukemia through various mechanisms (Table 2). m^6^A was reported to be reduced in about 70% of endometrial carcinoma cases. Endometrial cancer is associated with a METTL14 mutation (R298P), which results in decreased expression of the negative AKT regulator PHLPP2, increased expression of the positive AKT regulator mTORC2 and tumor cell proliferation [342]. Increased m^6^A by METTL14 suppresses renal cancer cell migration and invasion through down-regulation of P2RX6 protein translation and ATP-P2RX6-Ca^2+^-p-ERK1/2-MMP9 signaling [343]. METTL14 can suppress proliferation and metastasis of colorectal cancer and liver cancer through down-regulating oncogenic long non-coding RNA *XIST* and miRNAs, respectively [344,345]. METTL3 and METTL14 expression was reported to be downregulated in pediatric acute lymphoblastic leukemia (ALL) with ETV6/RUNX1 gene rearrangement [346]. However, paradoxically, METTL14 was reported to be highly expressed in normal hematopoietic stem/progenitor cells (HSPCs) and AML with t (11q23), t (15;17), or t (8;21) and necessary for development and maintenance of AML and self-renewal of leukemia stem/initiation cells (LSCs/LICs) [347]. Mechanistically, METTL14 exerts its oncogenic role by regulating the SPI1-METTL14-MYB/MYC signaling axis in myelopoiesis and leukemogenesis [347].

Like the m^6^A writer proteins METTL3 and METTL14, the m^6^A reader proteins YTHDF1 and YTHDF2 can act as either an oncogene or a tumor suppressor. YTHDF1 promotes cancer cell growth through upregulation of MYC [348] (Table 2). YTHDF1 also increases Wnt/β-catenin signaling and EIF3C-mediated mRNA translation, promoting tumor cell growth in colon cancer and ovarian cancer, respectively [349,350]. Furthermore, YTHDF1 inhibits neoantigen-specific immunity by binding to m^6^A-masked transcripts of lysosomal proteases. This binding increases lysosomal cathepsins translation, leading to inhibition of antigen presentation of wild-type dendritic cells and CD8^+^ T cell infiltration, and immune evasion by tumor cells in vivo [351]. YTHDF2 orchestrates epithelial-mesenchymal transition/proliferation dichotomy in pancreatic cancer cells [352] through YAP/TAZ signaling [353]. YTHDF2 regulates the lncRNA LINC00470/METTL3-mediated degradation of the tumor suppressor PTEN mRNA, which promotes cell growth and metastasis in gastric cancer [354]. YTHDC1 has been shown to promote m^6^A/METTL3-mediated tumor cell growth in glioblastoma (GBM) via regulation of nonsense-mediated mRNA decay (NMD) of serine- and arginine-rich splicing factors (SRSF) [355]. Tumor cell growth and metastasis is mediated by YTHDC2 through increased translation of c-Jun and ATF-2 in liver cancer [356], and HIF-1α in colon cancer [357]. However, reports have also shown that YTHDF1 and YTHDF2 can act as tumor suppressors. For example, YTHDF1 promotes the translation of m^6^A-marked HINT2 mRNA, a tumor suppressor, which inhibits ocular melanoma progression [358], and YTHDF2 suppresses cell proliferation and growth via destabilizing the EGFR mRNA in hepatocellular carcinoma [359].

In contrast to the plethora of studies on m^6^A writers and readers in cancer, less is known about the m^6^A demethylase, FTO. In breast cancer, FTO has been shown to enhance the PI3K/AKT signaling pathway and degrade BNIP3 (a mitochondrial pro-apoptotic protein) mRNA transcripts, promoting cancer cell growth [360,361]. Paradoxically, FTO can also suppress renal cancer cell growth by reducing m^6^A levels in its mRNA transcripts, which in turn increase PGC-1α expression [362].

#### 7.4.2. Nm and Its RMPs in Cancer

Nm is one of the most common RNA modifications and plays a role in the pathogenic processes of various human diseases [17]. Rare cases of non-small cell lung cancer (NSCLC) harboring CMTR1-ALK fusion have been reported, and do not respond to ALK inhibitor crizotinib [363]. Fibrillarin (FBL), the methyltransferase in the C/D box small nucleolar RNAs (SNORDs)/small nucleolar ribonucleoprotein complexes (snoRNPs), carries out Nm in rRNA [364]. FBL Nm promotion in rRNA increases internal ribosome entry site (IRES)-dependent translation initiation of key cancer genes and tumorigenesis. Meanwhile, p53 acts as a safeguard of protein synthesis by regulating FBL-mediated translation initiation [365]. Long noncoding RNA ZFAS1 was reported to promote small nucleolar RNA-mediated Nm in rRNA via NOP58 recruitment in colorectal cancer [366]. Increased Nm in 18S rRNA has been linked to the pathogenesis of AML [367,368]. Recently, HENMT1 was identified as the methyltransferase responsible for 3’-terminal Nm of mammalian miRNAs, which promotes 3’-terminal Nm in human non-small cell lung cancer (NSCLC) [369] (Table 2).

#### 7.4.3. RNA Editing and Its RMPs in Cancer

A-to-I editing and ADARs are known to contribute to tumorigenesis [370]. Like m^6^A and its RMPs, the roles of A-to-I editing and ADARs are also multifaceted. ADAR1 is overexpressed in many cancer types, including lung, liver and esophageal cancers and chronic myeloid leukemia (CML), and it acts predominantly as an oncogene. In contrast, ADAR2 expression is reduced in glioblastoma, in which ADAR2 acts as a tumor suppressor [371,372,373]. ADAR1 edits an intronic splicing silencer, leading to recruitment of SRSF7 and repression of exon inclusion, as well as cancer progression [374]. In antizyme inhibitor 1 (AZIN1), ADAR1-induced A-to-I editing causes the serine (S) → glycine (G) substitution at residue 367, leading to a cytoplasmic-to-nuclear translocation and “gain-of-function” phenotypes manifested by augmented tumor initiating potential and more aggressive cancer cell behavior [375]. ADAR1 promotes A-to-I editing in the 3’UTRs of ATM, GINS4 and POLH mRNA transcripts, resulting in increased expression of those oncogenic proteins and cancer cell proliferation in breast cancer [376]. Recently, ADAR1 was identified as a critical factor contributing to drug resistance in cancer cells, and the aberrant expression of ADAR1 promotes resistance to BET inhibitors in pancreatic cancer [377,378,379] (Table 2).

ADAR2 is essential for editing several miRNAs, such as miR-222/221 and the miR-21 precursor in vivo and in vitro, most of them oncogenic miRNAs [380]. However, its editing activity is impaired in glioblastoma cells, and restoring its activity inhibits tumor cell proliferation [380]. ADAR2 editing in tumor suppressor miRNAs, such as miR-589-3p and miRNA-379-5p, leads to increased expression of these tumor suppressor miRNAs [381,382]. Together, the A-to-I editing of these miRNAs by ADAR2 leads to inhibition of glioblastoma cell proliferation, migration and invasion (Table 2).

#### 7.4.4. m^5^C and Its RMPs in Cancer

RNA:m^5^C is mediated by NSUN RNA methyltransferase family proteins (Figure 2B) [51]. Upregulation of RNA:m^5^C and its writer proteins, especially NSUN1 and NSUN2, has been reported in a variety of solid tumors and leukemia and contributes to tumor cell proliferation, metastasis and drug resistance [55,57,331]. However, compared to a large body of studies on the pathophysiological function of m^6^A and its RMPs, much less is known about m^5^C and its RMPs.

NSUN1 (aka NOL1, NOP2 and p120) is a well-known tumor proliferation associated nucleolar protein [383,384] and its expression levels predict breast cancer disease progress and patient survival [385]. Our recent study demonstrates that NSUN1 expression is elevated in leukemia cells, and NSUN1 interacts with BRD4 to recruit elongating RNA polymerase II (eRNAPII) to nascent RNA, forming an active chromatin structure and transcription complex that mediates drug resistance in myeloid leukemia [127]. NSUN1 increases expression of oncogenic lncRNAs PVT1 and/or LINC00963 and inhibits expression of tumor suppressor miRNAs, leading to prostate cancer cell growth and metastasis [386,387]. Finally, NSUN1 can also interact with RPL6 to promote gallbladder cancer progression [388].

NSUN2 is responsible for the majority of mRNA:m^5^C in mammalian cells and is upregulated in various cancer types [57,389]. NSUN2 was found to be a downstream regulator of MYC and promotes cancer cell proliferation by RNAPIII methylation [390]. In bladder cancer, NSUN2 is involved in pathogenesis promotion through m^5^C deposition in the 3’ UTR of hepatoma-derived growth factor (HDGF). HDGF stabilizes the mRNA transcripts marked with m^5^C, which is selectively bound by YBX1 protein [391,392]. NSUN2-induced m^5^C in CDK1 mRNA promotes CDK1 translation, leading to tumor cell proliferation [392,393,394].

## 8. Conclusions and Future Directions

RNA modifications and RMPs are distributed widely in various subcellular organelles, such as nuclear particles, ribosomes and mitochondria, and identifying the role of RNA epigenetics in coordinating the functions of these subcellular organelles is likely to play an important part in more accurately characterizing multiple cellular functions and disease processes. It is increasingly clear that the role of RNA epigenetics is multifaceted and context-dependent in chromatin remodeling and gene expression. Although great progress has been made in elucidating the many roles of RNA epigenetics physiologically, the detailed mechanisms by which RNA modifications and RMPs control cancer progression and drug resistance remain largely unknown. There is still little known about how selective hyper- or hypo-regulation of RNA modifications in various RNA species contributes to tumor progression and therapeutic response/resistance. Great potential exists for the use of RNA modifications and RMPs as diagnostic/prognostic tools and therapeutic targets in future cancer therapy. However, given the large number of RNA modifications, an enormous amount of work lies ahead to fully understand the pathophysiological function of RNA modifications and their RMPs in cancer.

## Figures and Tables

**Figure 1 genes-12-00627-f001:**
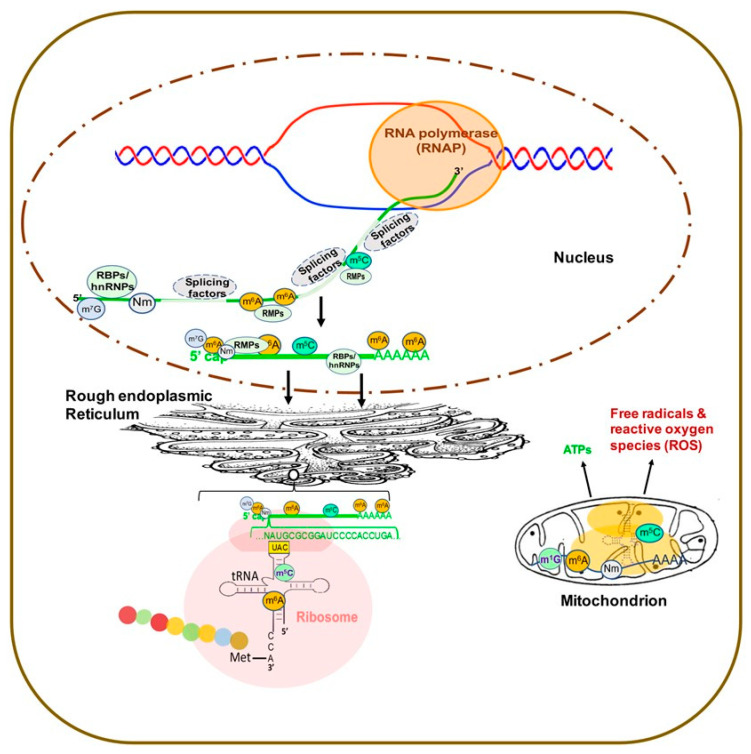
Subcellular Distribution of RNA Modifications and RMPs in Eukaryotes. Abbreviations: RMP, RNA modifying protein; RBP, RNA binding proteins; m^7^G, 7-methylguanosine; Nm, 2′-O-methylation; m^6^A, N6-methyladenosine; m^5^C, 5-methylcytosine; hnRNPs, heterogeneous nuclear ribonucleoproteins.

**Figure 2 genes-12-00627-f002:**
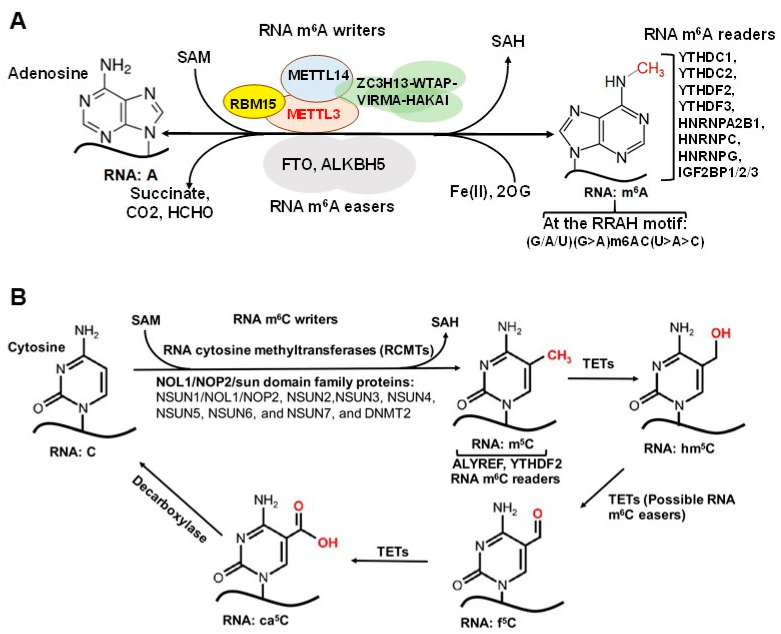
Formation, Recognition and Removal of RNA m^6^A and m^5^C in Eukaryotes. (**A**). Formation, Recognition and Removal of RNA:m^6^A. The METTL3/14 methyltransferase complex transfers methyl groups from SAM to N6-adenosines at the RRAH motifs in RNA. m^6^A is then recognized by m^6^A readers (m^6^A-selective binding proteins), and eventually removed by RNA m^6^A erasers. (**B**). Formation, Recognition and Removal of RNA:m^5^C. RNA m^5^C writers methylate cytosine residues, which are then recognized by m^5^C readers, or TETs, which oxidize m^5^C to hm^5^C, f^5^C and ca^5^C, respectively. Abbreviations: SAM, S-adenosylmethionine; SAH, S-adenosyl homocysteine; RBM15, RNA binding motif protein 15; METTL3/14, methyltransferase like 3/14; ZC3H13, zinc finger CCCH-type containing 13; WTAP, Wilms tumor suppressor gene WT1; VIRMA, Vir-like m6A methyltransferase associated; HAKAI, Cbl Proto-Oncogene Like 1; FTO, Fat mass and obesity associated; ALKBH5, AlkB homolog 5; YTHDC1/2, YTH domain containing 1/2; YTHDF2/3, YTH N6, methyladenosine RNA-binding protein 2/3; HNRNP family, heterogeneous nuclear ribonucleoproteins; IGF2BP1/2/3, insulin-like growth factor 2 mRNA binding protein 1/2/3. NSUN family, NOL1/NOP2/sun domain; DNMT2, DNA methyltransferase 2; ALYREF, Aly/REF export factor; YTHDF2, YTH N6-methyladenosine RNA binding protein 2; TETs, ten eleven translocation elements; hm^5^C, 5-hydroxymethylcytosine; f^5^C, 5-formylcytosine; ca^5^C, 5-carboxylcytosine.

**Figure 3 genes-12-00627-f003:**
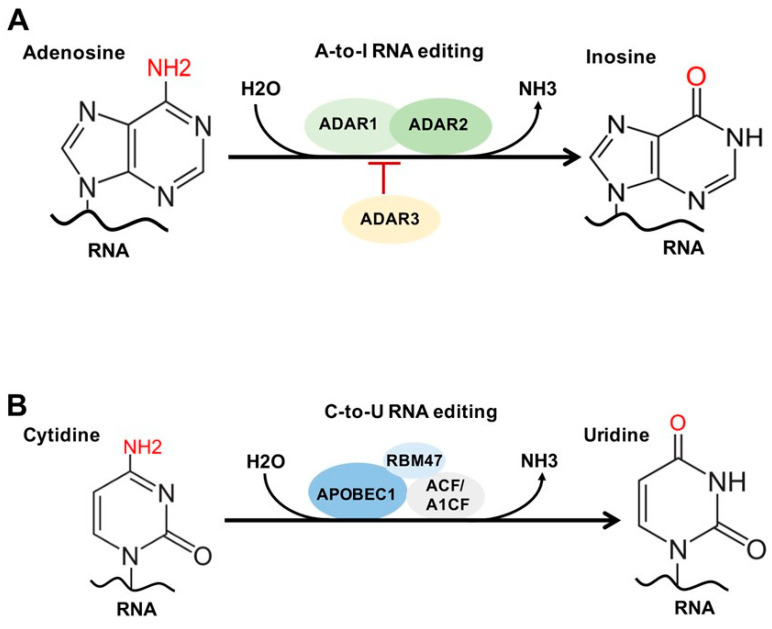
Molecular Reactions of RNA Adenosine to Inosine (A-to-I) and Cytidine to Uridine (C-to-U) Editing in Eukaryotes. (**A**). A-to-I RNA Editing Mechanism. ADAR1 and ADAR2 catalyze the site-specific conversion of A-to-I within imperfectly duplexed RNA. Meanwhile ADAR3 inhibits A-to-I editing. (**B**). C-to-U RNA Editing Mechanism. APOBEC1 and ACF bind to the RNA duplex, and RBM47 interacts with APOBEC1 and ACF, to produce C-to-U conversion via hydrolytic deamination of cytidine. Abbreviations: ADAR1/2, adenosine deaminases acting on RNA; APOBEC1, apolipoprotein B mRNA editing enzyme catalytic subunit 1; RBM47, RNA binding motif protein 47; ACF/A1CF, APOBEC1-complementation factor.

**Figure 4 genes-12-00627-f004:**
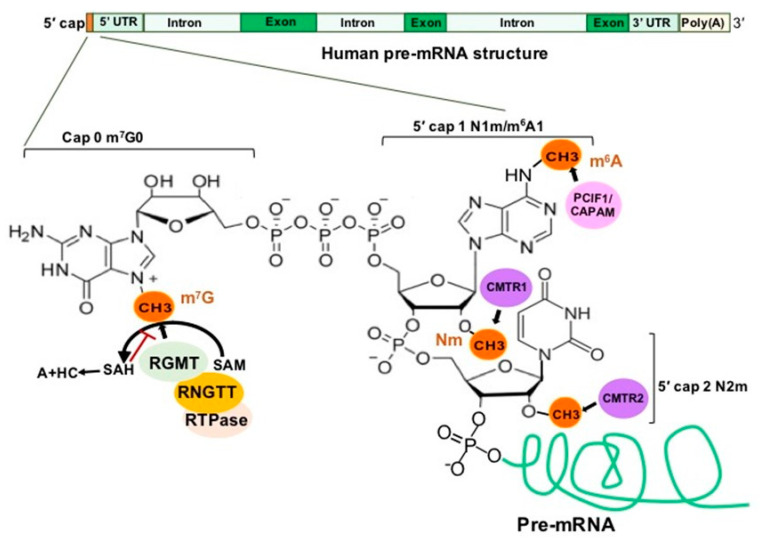
5′ Cap Structure of a Human Pre-mRNA and the Associated Methylations and Methyltransferase Complexes. Abbreviations: m^7^G, 7-methylguanosine; RGMT, RNA guanine methyltransferase; RNGTT, RNA guanylyltransferase and 5′ phosphatase; RTPase-RNA triphosphatase; SAM, S-adenosylmethionine; SAH, S-adenosyl homocysteine; PCIF1/CAPAM, PDX C-terminal inhibiting factor 1; CMTR1/2, cap methyltransferase 1/2.

**Figure 5 genes-12-00627-f005:**
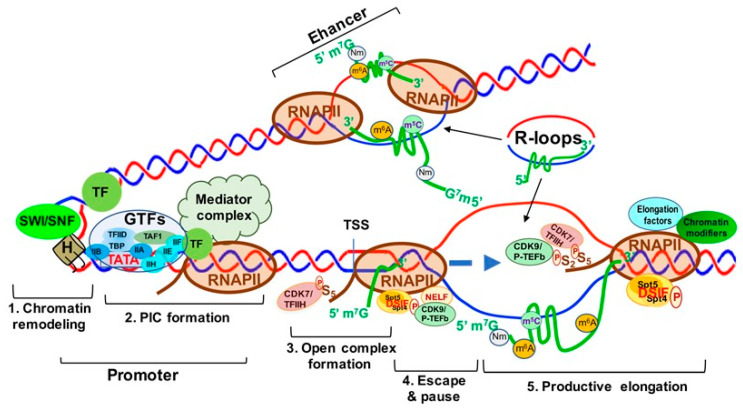
Transcriptionally Active Chromatin Structure and the Five Steps of Transcription Activation. **Step 1**. Chromatin remodeling; **Step 2**. Formation of transcription pre-initiation complex (PIC); **Step 3**. Unwinding of the DNA strands to form the open complex; **Step 4**. RNA polymerase II (RNAPII) pausing and escaping from the promoter; **Step 5**. Productive transcript elongation by RNAPII. Abbreviations: SWI-SNF, switch/Sucrose non-fermentable; TF, transcription factor; H, histone; PIC, preinitiation complex; GTFs, general transcription factors; TFIIA-F, transcription factor II A–F; TAF1, TATA-box binding protein associated factor 1; TBP, TATA-box binding protein; CDK7/TFIIH, cyclin-dependent kinase 7; TSS, transcription start Site; Spt5/4, suppressor of Ty 5 and 4 (transcription elongation factors 5 and 4); DSIF, DRB sensitivity-inducing factor; CDK9/p-TEFb, cyclin-dependent kinase 9.

**Figure 6 genes-12-00627-f006:**
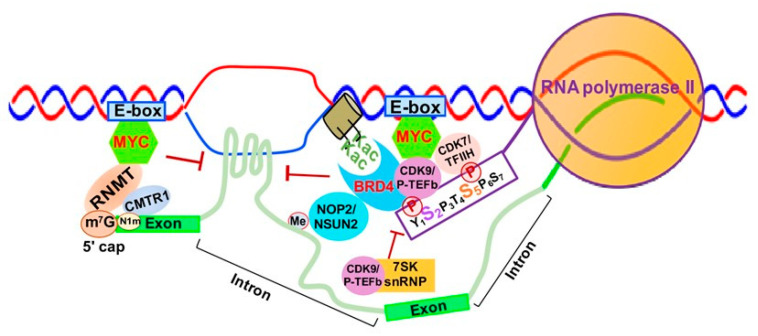
MYC-mediated RNA 5′ Capping, Transcription Elongation and Chromatin Structure. Abbreviations: E-box- enhancer box; MYC- MYC proto-oncogene BHLH transcription factor; RNMT, RNA methyltransferase; CMTR1, cap methyltransferase 1; K-ac, acetylated lysine; CDK9/P-TEFb, cyclin-dependent Kinase 9; CDK7/TFIIH, Cyclin-dependent Kinase 7; BRD4, Bromodomain Containing 4; 7SK snRNP, 7SK small nuclear ribonucleoprotein.

**Figure 7 genes-12-00627-f007:**
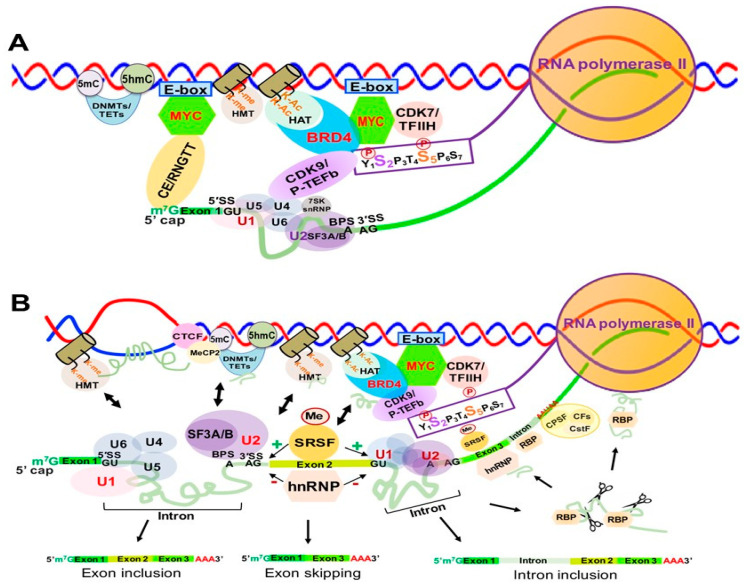
Transcription-associated Spliceosome Assembly and Alternative Splicing in Eukaryotes. (**A**). Determination of the first exon-intron boundary and formation of initial spliceosome on pre-mRNA. (**B**). Transcription-associated determination of the internal exon-intron boundaries, spliceosome assembly and alternative splicing of pre-mRNA. Abbreviations: CTCF, CCCTC binding protein; MeCP2, methyl-CpG binding protein 2; DNMT, DNA methyltransferase; TET, ten eleven translocation elements; E-box, enhancer box; CE/RNGTT, RNA guanylyltransferase and 5′-Phosphatase; HMT, histone methyltransferase; HAT, histone acetyltransferase; BRD4, bromodomain containing 4; U1-U6, small nuclear ribonucleoprotein U1-U6 subunit; CDK9/P-TEFb, cyclin-dependent kinase 9; 7SK snRNP, 7SK small nuclear ribonucleoprotein; SF3A/B, splicing factor 3a/b subunit; CDK7/TFIIH, cyclin-dependent kinase 7; TETs, ten eleven translocation elements; SRSF, serine/arginine-rich splicing factors; hnRNP, heterogeneous nuclear ribonucleoproteins; RBP, RNA binding proteins.

**Figure 8 genes-12-00627-f008:**
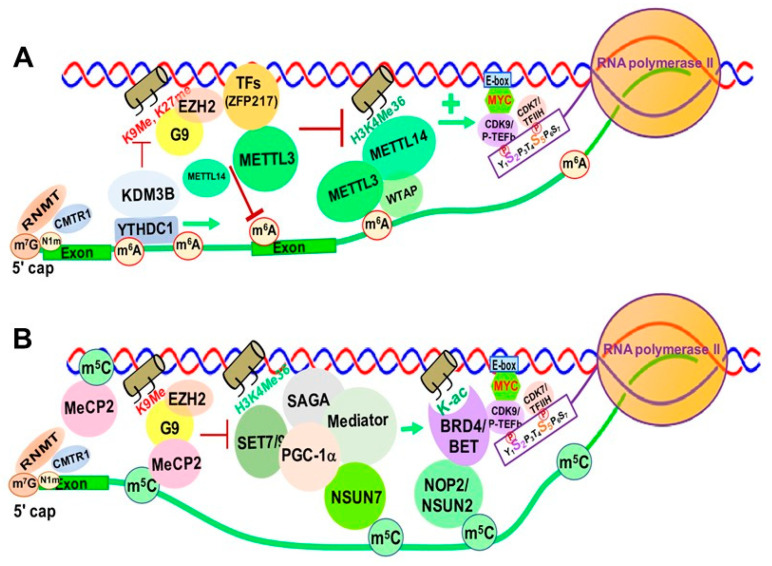
RNA Modification/RMP-mediated, Transcription-associated Chromatin Structural Changes in Mammalian Cells. (**A**). m^6^A and its RMPs-mediated chromatin structural changes. (**B**). m^5^C and its RMPs-mediated chromatin structural changes. Abbreviations: m^6^A, N6-methyladenine; RNMT, RNA Methyltransferase; CMTR1, cap methyltransferase 1; KDM3B, lysine demethylase 3B; YTHDC1, YTH Domain Containing 1; EZH2, enhancer of Zeste 2 Polycomb repressive Complex 2 Subunit; METTL3/14, methyltransferase Like 3/14; K9me, lysine 9 methyl; K27me, lysine 27 methyl; TFs, Transcription Factors; ZFP217, zinc finger Protein 217; H3K4me36, histone 3 lysine 36 tri-methylation; E-box, enhancer box; CDK9/P-TEFb, cyclin-dependent kinase 9; CDK7/TFIIH, cyclin-dependent kinase 7. m^5^C, 5-methylcytosine; MeCP2, methyl-CpG binding protein 2; SET7/9, SET Domain containing 7/9 histone lysine methyltransferase; SAGA, Spt-Ada-Gcn5-acetyltransferase; PGC-1α, peroxisome proliferator-activated receptor γ coactivator 1-α; NSUN7, NOP2/Sun RNA methyltransferase member 7; H3K4me36, histone 3 lysine 36 tri-methylation; NOP2/NSUN1, NOP2 nucleolar protein/NOP2/Sun RNA methyltransferase 1; K-ac, lysine acetylation; BRD5/BET, bromodomain containing 4; E-box, enhancer box; CDK9/pTEFb, cyclin-dependent kinase 9; CDK7/TFIIH, cyclin-dependent Kinase 7; MYC, MYC proto-oncogene BHLH transcription factor.

**Table 1 genes-12-00627-t001:** The types and functions of regulatory non-coding RNAs (ncRNAs) in mammalian cells.

Types of ncRNAs	NcRNA Abbreviation	ncRNA Strands	Size of ncRNAs	Function	References
Small ncRNAs			<200 nucleotides		
Small interfering RNAs	siRNAs	double	21–23	suppress homology containing transcripts via mRNA degradation	[179,180]
Micro-RNAs	miRNAs	single	~22	repress translation; accelerate mRNA degradation	[181]
Piwi-interacting RNAs	piRNAs	single	21–25	regulate gene expression; fight viral infection	[182,183]
Small nucleolar RNAs	snoRNAs	single	60–300	rRNA, snRNA, and other RNA posttranscriptional modification	[182,183]
Small Cajal body RNAs (snoRNA)	scaRNAs	single	60–300	biochemical modification of premature spliceosomal RNAs	[182,183]
Long non-coding RNAs	lncRNAs	single	>200 nucleotides	Chromatin remodeling; transcriptional and post-transcriptional regulation	[184]

**Table 2 genes-12-00627-t002:** Implications of common RNA modifications and RNA modifying proteins in human diseases.

Disease Categories/ Subcategories	RNA Modifications & RMPs	Possible Pathogenic Mechanisms	References
**Neuronal Systems:** 1. Neurogenesis defects & neural degenerative disorders. 2. Memory defects 3. Psychiatric Diseases 4. Intellectual Disability	**M^6^A:**		
	METTL3/14	Knockout (KO) of *Mettl3/14* → ↑ cell cycle and ↓ NSC proliferation via histone modifications	[268,269,270,271]
	YTHDF1	↑ Translation of axon guidance protein Robo3.1	[272]
	FTO	FTO regulates *Pdfra/Socs5-Stat3*; TSC1-mTOR-Tau signaling	[273,274]
	**RNA editing:**		
	ADARs	Affect mRNA abundance and miRNA binding→ regulating development and function of neurons and neurotransmitters	[283,284]
	**Nm:**		
	FTSJ1, CMTR1	FTSJ1 affects tRNA:Nm and neurotransmitter translationCMTR1 affects 5′ cap RNA: Nm and translation	[286,287,288,289]
	**M^5^C:**		
	NSUN2	Affects tRNA:m^5^C and protein translation	[291,292,293,294,295,296]
**Metabolic Disorders:** 1 Type 2 Diabetes (T2D) 2. Obesity	**M6A:**		
	METTL3,	↑METTL3/14/WTAP (↑ insulin-IGF1-AKT-PDX1 pathway(↑ β-cell survival(glucose homeostasis	[297,298,299]
	METTL14	↑METTL3/14/WTAP (↑MCE (↑ adipogenesis	[300]
	YTHDF2	YTHDF2 →↓ ZFP217-induced MCE → ↓ adipogenesis	[301]
	FTO	↑ FTO→ ↑ adipogenic RUNX1T1; ↓Wnt/β-catenin; ↑ATG5/7→ adipogenesis	[302,303,304,305,306,307]
**Immune System:** 1. Innate immunity, autoimmune & viral infections 2. Adaptive immunity	**M6A:**		
	METTL3	↓METTL3/14 or YTHDF2( ↑ IFNB1 mRNA	[308,309,310,311,312,313,314,315,316,317,318]
	METTL14	↓METTL3( ↓Rev-RRE binding( ↓viral RNA export & production	
	YTHDF2	↓METTL3( ↓DC activation; ↓ IL7/2-STAT5-SOCS	
	**A-to-I editing**		
	ADAR1/p150	↓ADAR1→ ↑dsRNA sensor-triggered INF responses → autoimmune and cell death	[319,320,321]
	**5′ cap Nm:**		
	CMTR1	↑5′cap Nm→ ↓viral mRNA binding by sensor Mda5 & IFIT1→ ↓INF	[21,322,323,324,325,326]
**Cancer & Leukemia** 1. Tumorigenesis, cell proliferation, and metastasis 2. Immune evasion and immunotherapy 3. Drug Resistance **Cancer & Leukemia1** 1. Tumorigenesis, cell proliferation, and metastasis 2. Immune evasion and immunotherapy 3. Drug Resistance	**M6A:** METTL3 YTHDF1 YTHDF2 YTHDC1 YTHDC2 FTO	**Acting as oncogene:**	
		↑METTL3( ↑oncogenic mRNA & miRNAs; ↑MYC; ↑Wnt/β-catenin signaling	[332,333,334,335,336,337,338,339,340,341,342,343]
		↑YTHDF1( ↑EiF3C translation; ↑Wnt/β-catenin; ↓ antigen presentation by DCs( immune evasion	[348,349,350,351,352]
		↑YTHDF2( ↑oncogenic lncRNA; ↑YAP signalng; ↑YTHDC( ↑translation of c-JUN, ATF-2, HIF-1a	[353,354,355,356,357]
		↑ FTO( ↑ PI3K/AKT; ↓ BNIP3 mRNA	[360,361]
	METTL14 YTHDF1 YTHDF2 FTO	**Acting as tumor suppressor:**	
		↑METTL14 ↑ (↑tumor suppressor miR-375/Yes-associated protein 1 (YAP1) pathway; ↓lncRNA XIST; ↓ERK9/MMP9 signaling; ↓oncogenic miRNAs	[339,344,345,346,347]
		↑YTHDF1( ↑ tumor suppressor HINT2; ↑YTHDF2( Destabilizing EGFR mRNA	[358,359]
		↑ FTO(↑PGC-1α signaling	[362]
	**Nm:** Fibrillarin HENMT1	**Acting as oncogene:**	
		↑Fibrillarin(↑ rRNA:Nm(↑translation of oncogenic proteins	[365]
		↑HENMT1(stablizing miR-21-5p(↓programed cell death protein 4	[369]
	**RNA editing:** ADAR1/p150 ADAR2	**Acting as oncogene:**	
		↑ADAR1(↑oncogenic miRNAs; ↑SRSF7-mediated intron inclusion; “gain-of-function” mutation in AZIN1	[371,372,373,374,375]
		↑ADAR1(A-to-I editing in 3′UTRs of ATM, GINS4 and POLH Mrna	[376]
		↑ADAR1( ↑ resistance to BET inhibitors & checkpoint blockade	[377,378,379]
		**Acting as tumor suppressor:** ↑ ADAR2( ↑oncogenic miRNAs; ↑ tumor suppressor miRNAs	[380,381,382]
	**M^5^C:** NSUN1 NSUN2	**Acting as tumor suppressor:**	
		↑NSUN1(↑BRD4-mediated eRNAPII recruitment to pre-mRNA	[127]
		↑NSUN1(↑oncogenic mRNA & miRNAs	[383,384,385,386,387,388]
		↑NSUN2(↑MYC-mediated RNAPIII transcripts; ↑mRNA stability	[389,390,391,392,393,394]

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
