# Peer review of "RNA Epigenetics: Fine-Tuning Chromatin Plasticity and Transcriptional Regulation, and the Implications in Human Diseases"

_genes, 2021, doi:10.3390/genes12050627_

Round 1
Reviewer 1 Report
This is an interesting review focusing on RNA epigenetics an, gene expression and human desease.
The manuscript is overall well written.
I think it will be a reference review for people working in the field.
Author Response
Reviewer 1:
Open Review
( ) I would not like to sign my review report
(x) I would like to sign my review report
English language and style
( ) Extensive editing of English language and style required
( ) Moderate English changes required
(x) English language and style are fine/minor spell check required
( ) I don't feel qualified to judge about the English language and style
Is the work a significant contribution to the field? 4 |
|
Is the work well organized and comprehensively described? 4 |
|
Is the work scientifically sound and not misleading? 3 |
|
Are there appropriate and adequate references to related and previous work? 4 |
|
Is the English used correct and readable? 4 |
Comments and Suggestions for Authors
This is an interesting review focusing on RNA epigenetics an, gene expression and human desease.
The manuscript is overall well written.
I think it will be a reference review for people working in the field.
ANS: We want to thank the review 1 for his/her time and the comments. We have re-checked our manuscript and the cited references to make sure their scientific accuracy.
Reviewer 2 Report
The manuscript entitled "RNA Epigenetics: Fine-Tuning Chromatin Plasticity and Gene Expression, and the Implications in Human Diseases." describes a role of RNA modifications and RMPs in transcription.
The manuscript is clearly written and the grammar and spelling within the manuscript are pretty good. I enjoyed reviewing the manuscript and would recommend it for publication. While a number of ncRNAs & RNA modifications studies in mammals have recently come out in this area, I believe that this review article can still significantly contribute to summarize our limited knowledge in this field. My comments can be found below.
I think that the authors should re-draw all of the figures and Tables. For me, they are so busy figures and tables. It is very difficult to understand all of them.
Author Response
Reviewer 2:
Open Review
(x) I would not like to sign my review report
( ) I would like to sign my review report
English language and style
( ) Extensive editing of English language and style required
(x) Moderate English changes required
( ) English language and style are fine/minor spell check required
( ) I don't feel qualified to judge about the English language and style
Is the work a significant contribution to the field? 4 |
|
Is the work well organized and comprehensively described? 4 |
|
Is the work scientifically sound and not misleading? 4 |
|
Are there appropriate and adequate references to related and previous work? 4 |
|
Is the English used correct and readable? 4 |
Comments and Suggestions for Authors
The manuscript entitled "RNA Epigenetics: Fine-Tuning Chromatin Plasticity and Gene Expression, and the Implications in Human Diseases." describes a role of RNA modifications and RMPs in transcription.
The manuscript is clearly written and the grammar and spelling within the manuscript are pretty good. I enjoyed reviewing the manuscript and would recommend it for publication. While a number of ncRNAs & RNA modifications studies in mammals have recently come out in this area, I believe that this review article can still significantly contribute to summarize our limited knowledge in this field. My comments can be found below.
I think that the authors should re-draw all of the figures and Tables. For me, they are so busy figures and tables. It is very difficult to understand all of them.
ANS: We want to thank the review 2 for his/her time and the comments. Regarding the complexity of the figures and tables, we got very different opinions from other experts in epigenetics and transcription, who suggested to add more details to those figures and tables. I think that it is difficult to achieve the maximum simplicity without loss of the important information in those figures. However, we appreciate the reviewer’s opinion and try to revise the figures to eliminate as much non-essential details as possible. Thanks again.
Reviewer 3 Report
This is an in-depth literature review of epitranscriptomics, well written and referenced. This will provide the field with very precise and well-documented references.
The term chromatin structure is used casually throughout the manuscript, chromatin structure according to chromatin biologists includes an array of nucleosomes and well-defined nucleosome depleted regions.
In most of the figures, nucleosomes are not well depicted and proper weightage is not given to them.
Figures and legends not edited well "$" appears randomly in figures mostly editing issue needs to be taken care of.
Figure legends appear like an expansion of abbreviations at many places.
The text has some editing issues randomly seen bolded at places that need attention.
Author Response
Reviewer 3:
Open Review
(x) I would not like to sign my review report
( ) I would like to sign my review report
English language and style
( ) Extensive editing of English language and style required
(x) Moderate English changes required
( ) English language and style are fine/minor spell check required
( ) I don't feel qualified to judge about the English language and style
Is the work a significant contribution to the field? 3 |
|
Is the work well organized and comprehensively described? 3 |
|
Is the work scientifically sound and not misleading? 4 |
|
Are there appropriate and adequate references to related and previous work? 4 |
|
Is the English used correct and readable? 5 |
Comments and Suggestions for Authors
This is an in-depth literature review of epitranscriptomics, well written and referenced. This will provide the field with very precise and well-documented references.
The term chromatin structure is used casually throughout the manuscript, chromatin structure according to chromatin biologists includes an array of nucleosomes and well-defined nucleosome depleted regions.
In most of the figures, nucleosomes are not well depicted and proper weightage is not given to them.
Figures and legends not edited well "$" appears randomly in figures mostly editing issue needs to be taken care of.
Figure legends appear like an expansion of abbreviations at many places.
The text has some editing issues randomly seen bolded at places that need attention.
ANS: We want to thank the review 3 for his/her time and the comments. Regarding the terms of chromatin and nucleosomes used in this review, we purposely focus on the recent proceeding in RNA and its modifications-mediated active chromatin or transcriptionally active chromatin because RNA modifications and splicing occur co-transcriptionally and have the most profound and direct effects on transcriptionally active chromatin. Moreover, to the best of our knowledge, this important field has not been much reviewed in the literatures. By the way, the corresponding author on this manuscript was trained in one of the top yeast transcription laboratories and has well understanding of chromatin and nucleosome and their role in transcriptional regulation. For the original concept of “active chromatin”, please refer to the references: Weisbrod S. Active chromatin. Nature. 1982; 297(5864): 289-95. Reeves R. Transcriptionally active chromatin. Biochim Biophys Acta. 1984; 782 (4): 343-93. However, the last three decades of research has demonstrated the importance of non-coding RNA and its modifications in shaping chromatin structure. The early concept of active chromatin evolved into what current perspective of RNA epigenetics-mediated chromatin structural changes. Again, we appreciate the review’s input very much, and therefore, we have changed the title of this manuscript to "RNA Epigenetics: Fine-Tuning Chromatin Plasticity and Transcriptional Activation, and the Implications in Human Diseases."
Regarding the typos and errors in the figures, figure legends and text, we have made the corrections. We want to thank the reviewer for thorough review on our manuscript.